# Characterizing Forbidden Pairs for the Edge-Connectivity of a Connected Graph to Be Its Minimum Degree

Junfeng Du [1,2], Ziwen Huang [3] and Liming Xiong [4,*]

1 School of Mathematics and Statistics, Beijing Institute of Technology, Beijing 100081, China; dujf1990@163.com
2 Department of Mathematics, Beijing University of Chemical Technology, Beijing 100029, China
3 School of Mathematics and Computer Science, Center of Applied Mathematics, Yichun University, Yichun 336000, China; zwhuang@aliyun.com
4 School of Mathematics and Statistics, Beijing Key Laboratory on MCAACI, Beijing Institute of Technology, Beijing 100081, China
* Correspondence: lmxiong@bit.edu.cn

**Abstract:** Let $\mathcal{H}$ be a class of given graphs. A graph $G$ is said to be $\mathcal{H}$-free if $G$ contains no induced copies of $H$ for any $H \in \mathcal{H}$. In this article, we characterize all connected subgraph pairs $\{R, S\}$ guranteeing the edge-connectivity of a connected $\{R, S\}$-free graph to have the same minimum degree. Our result is a supplement of Wang et al. Furthermore, we obtain a relationship of forbidden sets when those general parameters have the recurrence relation.

**Keywords:** forbidden subgraph; edge-connectivity; minimum degree; parameter





## 1. Introduction

In this paper, we consider finite simple graphs only. For terminology and notation not defined here, we refer the readers to Bondy and Murty [1].

Let $G$ be a connected graph with vertex set $V(G)$ and edge set $E(G)$. Therefore, $n(G) = |V(G)|, e(G) = |E(G)|, \kappa(G), \kappa'(G)$ and $\delta(G)$ mean the *order, size, connectivity, edge-connectivity* and *minimum degree* of $G$, respectively. Suppose $u$ is a vertex of $G$. Then $N_G(u)$ denotes $\{x : ux \in E(G)\}$, which is also called the *neighbors* of $u$ in the graph $G$. Let $S \subseteq V(G)$ (or $E(G)$ respectively). The subgraph of $G$ induced by $S$ is denoted by $G[S]$, vertex induced subgraph and edge induced subgrph. Furthermore, we use $G - S$ to denote the subgraph $G[V(G)\backslash S]$ (or $G[E(G)\backslash S]$ respectively). The *distance* between two vertices $x, y \in V(G)$, denoted by $d_G(x, y)$, is the length of a shortest path between the two vertices $x$ and $y$, while the *diameter* of a graph $G$, denoted by $diam(G)$, is the greatest distance between any pair of vertices of $G$.

Let $H$ be a given graph. A graph $G$ is said to be *H-free* if any induced subgraph of $G$ is not isomorphic to $H$. If $G$ is $H$-free, then $H$ is called a *forbidden subgraph* of $G$. Note that if $H_1$ is an induced subgraph of $H_2$, then every $H_1$-free graph is also $H_2$-free. Let $\mathcal{H}$ be a set of connected graphs, the graph $G$ is $\mathcal{H}$-free if $G$ is $H$-free for every $H \in \mathcal{H}$. For two sets $\mathcal{H}_1$ and $\mathcal{H}_2$ of connected graphs, we write $\mathcal{H}_1 \preceq \mathcal{H}_2$ which means that for every graph $H_2 \in \mathcal{H}_2$, there exists a graph $H_1 \in \mathcal{H}_1$ such that $H_1$ is an induced subgraph of $H_2$. By the definition, we know that if $\mathcal{H}_1 \preceq \mathcal{H}_2$, then clearly every $\mathcal{H}_1$-free graph is also $\mathcal{H}_2$-free.

It always means that we use $K_n$, $K_{s,t}$ to denote the complete graph of order $n$, and the complete bipartite graph with partition sets of size $s$ and $t$, respectively. So the $K_1$ is a trivial graph, $K_3$ is a triangle, $K_{1,r}$ is a star (the $K_{1,3}$ is also called a *claw*). A clique $C$ is a subgraph of a graph $G$ such that $G[V(C)]$ is a complete graph, and the *clique number* $\omega(G)$ of a graph $G$ is the maximum cardinality of a clique of $G$. Then we will show some special graphs which are needed: (see Figure 1)

- $P_i$, the path with order $i$, hence $P_1 = K_1$ and $P_2 = K_2$;
- $Z_i$, a graph obtained by identifying a vertex of a $K_3$ with an end-vertex of a $P_{i+1}$;
- $H_1$, a graph obtained by identifying a vertex of a $K_3$ with the one-degree vertex of a $Z_1$;
- $T_{i,j,k}$, a graph consisting of three paths $P_{i+1}, P_{j+1}$ and $P_{k+1}$ with the common starting vertex.

Let $X$ and $Y$ be nonempty subsets of $V(G)$. We denote by $E[X, Y]$ the set of edges of $G$ with one end in $X$ and the other end in $Y$, and by $e(X, Y)$ their number. When $Y = V(G) \backslash X$, the set $E[X, Y]$ is called the *edge cut* of $G$ associated with $X$. An edge cut set $S$ with the minimum number of edges is called a *minimum edge cut*. It is well-known that $\kappa(G) \leq \kappa'(G) \leq \delta(G)$. In [2], Wang, Tsuchiya and Xiong characterize all the pairs $R, S$ such that every connected $\{R, S\}$-free graph $G$ has $\kappa(G) = \kappa'(G)$.

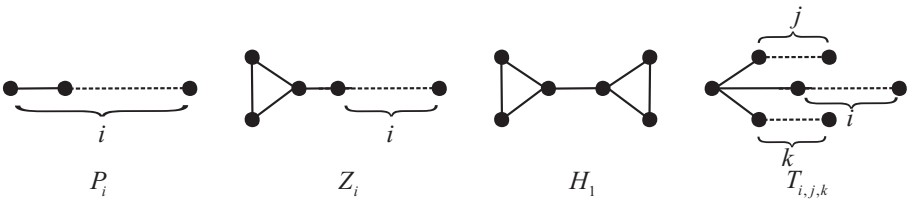

**Figure 1.** Some special graphs: $P_i, Z_i, H_1$ and $T_{i,j,k}$.

**Theorem 1** (Wang et al. [2])**.** *Let $S$ be a connected graph. Then $G$ is a connected $S$-free graph which implies $\kappa(G) = \kappa'(G)$ if and only if $S$ is an induced subgraph of $P_3$.*

**Theorem 2** (Wang et al. [2])**.** *Let $\mathcal{H} = \{R, S\}$ be a set of two connected graphs such that $R, S \neq P_3$. Then $G$ is a connected $\mathcal{H}$-free graph that $\kappa(G) = \kappa'(G)$ if and only if $\mathcal{H} \preceq \{Z_1, P_5\}$, $\mathcal{H} \preceq \{Z_1, K_{1,4}\}$, $\mathcal{H} \preceq \{Z_1, T_{1,1,2}\}$, $\mathcal{H} \preceq \{P_4, H_0\}$ or $\mathcal{H} \preceq \{K_{1,3}, H_0\}$, where $H_0$ is the unique simple graph with degree sequence $4, 2, 2, 2, 2$, i.e., which can be obtained from $H_1$ by the contracted edge whose endvertices are of degree 3.*

In [3], Hellwig and Volkmann introduce several sufficient conditions for $\kappa'(G) = \delta(G)$.

**Theorem 3.** *Let $G$ be a connected graph satisfying one of the following conditions:*

1. *(Chartrand [4]) $n(G) \leq 2\delta(G) + 1$,*
2. *(Lesniak [5]) $d_G(u) + d_G(v) \geq n(G) - 1$ for all pairs $u, v$ of nonadjacent vertices,*
3. *(Plesník [6]) $diam(G) = 2$,*
4. *(Volkmann [7]) $G$ is bipartite and $n(G) \leq 4\delta(G) - 1$,*
5. *(Plesník and Znám [8]) there are no four vertices $u_1, u_2, v_1, v_2$ with $d_G(u_1, u_2), d_G(u_1, v_2), d_G(v_1, u_2), d_G(v_1, v_2) \geq 3$,*
6. *(Plesník and Znám [8]) $G$ is bipartite and $diam(G) = 3$,*
7. *(Xu [9]) there exist $\lfloor n(G)/2 \rfloor$ pairs $(u_i, v_i)$ of vertices such that $d_G(u_i) + d_G(v_i) \geq n(G)$ for $i = 1, 2, \cdots, \lfloor n(G)/2 \rfloor$,*
8. *(Dankelmann and Volkmann [10]) $\omega(G) \leq p$ and $n(G) \leq 2\lfloor p\delta(G)/(p-1) \rfloor - 1$.*

*Then $\kappa'(G) = \delta(G)$.*

## 2. Our Results

In this paper, we characterize all forbidden subgraphs sets $\mathcal{H}$ of graphs such that every connected $\mathcal{H}$-free graph implies $\kappa'(G) = \delta(G)$ for $|\mathcal{H}| = 1, 2$.

**Theorem 4.** *Let $S$ be a connected graph. Then $G$ being a connected $S$-free graph implies $\kappa'(G) = \delta(G)$ if and only if $S$ is an induced subgraph of $P_4$.*

**Theorem 5.** *Let $\mathcal{H} = \{R, S\}$ be a set of two connected graphs such that $R$ and $S$ are not an induced subgraph of $P_4$. Then $G$ being a connected $\mathcal{H}$-free graph implies $\kappa'(G) = \delta(G)$ if and only if $\mathcal{H} \preceq \{H_1, P_5\}$, $\mathcal{H} \preceq \{Z_2, P_6\}$, or $\mathcal{H} \preceq \{Z_2, T_{1,1,3}\}$.*

Note that whenever a connected graph $G$ satisfies $\kappa(G) < \kappa'(G)$ or $\kappa'(G) < \delta(G)$, it satisfies $\kappa(G) < \delta(G)$. Then we want to characterize the forbidden subgraphs for $\kappa(G) = \delta(G)$.

In fact, for $f(G), g(G), t(G)$ are three invariants of $G$ with $f(G) \leq g(G) \leq t(G)$, we also present a general result which may help us to deal with the relationship between them. In order to state the result clearly, we further introduce some notations. For two sets of given graphs $\mathcal{H}_1 = \{H_1^1, H_1^2, \cdots, H_1^n\}$ and $\mathcal{H}_2 = \{H_2^1, H_2^2, \cdots, H_2^n\}$, we use $\mathcal{H}_1 \underset{ind}{\bigcap} \mathcal{H}_2$ to denote the set with order n, in which each element is the common induced subgraph of one graph in $\mathcal{H}_1$ and one graph in $\mathcal{H}_2$, respectively, i.e., $\mathcal{H}_1 \underset{ind}{\bigcap} \mathcal{H}_2 := \{S_1, S_2, \cdots, S_n | S_i$ is the common induced subgraph of $H_1^j$ and $H_2^k, i, j, k \in \{1, 2, \cdots, n\}\}$.

Now we may get the following result (A similar result by replacing the parameter with a set of subgraphs; you may see [11], in its last section).

**Theorem 6.** *Let $G$ be a connected graph, and $f(G), g(G), t(G)$ are three invariants of $G$ with $f(G) \leq g(G) \leq t(G)$. If the following statements hold:*

1. *$G$ is $\mathcal{H}$-free implies $f(G) = g(G)$ if and only if $\mathcal{H} \in \mathbf{H_1}$;*
2. *$G$ is $\mathcal{H}$-free implies $g(G) = t(G)$ if and only if $\mathcal{H} \in \mathbf{H_2}$,*

*then $G$ is $\mathcal{H}$-free implies $f(G) = t(G)$ if and only if $\mathcal{H} \in \mathbf{H_1} \bigcap \mathbf{H_2}$.*

**Proof.** First suppose $G$ is $\mathcal{H}$-free and $\mathcal{H} \in \mathbf{H_1} \bigcap \mathbf{H_2}$, then $\mathcal{H} = \mathcal{H}_1 \underset{ind}{\bigcap} \mathcal{H}_2$ for some $\mathcal{H}_1 \in \mathbf{H_1}$ and $\mathcal{H}_2 \in \mathbf{H_2}$. By the definition of $\mathcal{H}_1 \underset{ind}{\bigcap} \mathcal{H}_2$, we can see that $\mathcal{H} \underset{ind}{\preceq} \mathcal{H}_1$ and $\mathcal{H} \underset{ind}{\preceq} \mathcal{H}_2$. Therefore, $G$ is $\mathcal{H}_1$-free and $\mathcal{H}_2$-free. By (1) and (2), $f(G) = g(G)$ and $g(G) = t(G)$. It means that $f(G) = g(G) = t(G)$. This completes the sufficiency.

Now we prove the necessity. Suppose that $f(G) = t(G)$. Then $f(G) = g(G) = t(G)$ since $f(G) \leq g(G) \leq t(G)$. By the necessity of (1) and (2), $G$ must be $\mathcal{H}_i$-free for each $\mathcal{H}_i \in \mathbf{H_i}$, $i = 1, 2$. Therefore, $G$ should be $\mathcal{H}_1 \bigcap \mathcal{H}_2$-free. By the definition of $\mathbf{H_1} \bigcap \mathbf{H_2}$, $\mathcal{H} = \mathcal{H}_1 \underset{ind}{\bigcap} \mathcal{H}_2 \in \mathbf{H_1} \bigcap \mathbf{H_2}$. This completes the proof. □

By Theorems 1, 4 and 6, we can obtain the following corollary:

**Corollary 1.** *Let $S$ be a connected graph. Then $G$ being a connected $S$-free graph implies $\kappa(G) = \delta(G)$ if and only if $S$ is an induced subgraph of $P_3$.*

Moreover, note that "$G$ is $P_4$-free" implies "$\kappa'(G) = \delta(G)$" (see Theorems 4), this also means that if $G$ is $\{P_4, S\}$-free, then $\kappa'(G) = \delta(G)$, here $S$ can be any subgraph of $G$. Therefore, by Theorems 2, 5 and 6, we also can obtain the following corollary:

**Corollary 2.** *Let $\mathcal{H} = \{R, S\}$ be a set of two connected graphs such that $R$ and $S$ are not an induced subgraph of $P_3$. Then $G$ being a connected $\mathcal{H}$-free graph implies $\kappa(G) = \delta(G)$ if and only if $\mathcal{H} \preceq \{H_0, P_4\}$, $\mathcal{H} \preceq \{Z_1, P_5\}$, or $\mathcal{H} \preceq \{Z_1, T_{1,1,2}\}$.*

### 3. The Necessity Part of Main Results

We first construct some families of connected graphs $\mathcal{G}_i, i = 1, \cdots, 7$ (see Figure 2. Here $m_i \geq 4$). It is easy to see that each $G \in \mathcal{G}_i$ satisfies $1 = \kappa'(G) < \delta(G) = 2$.

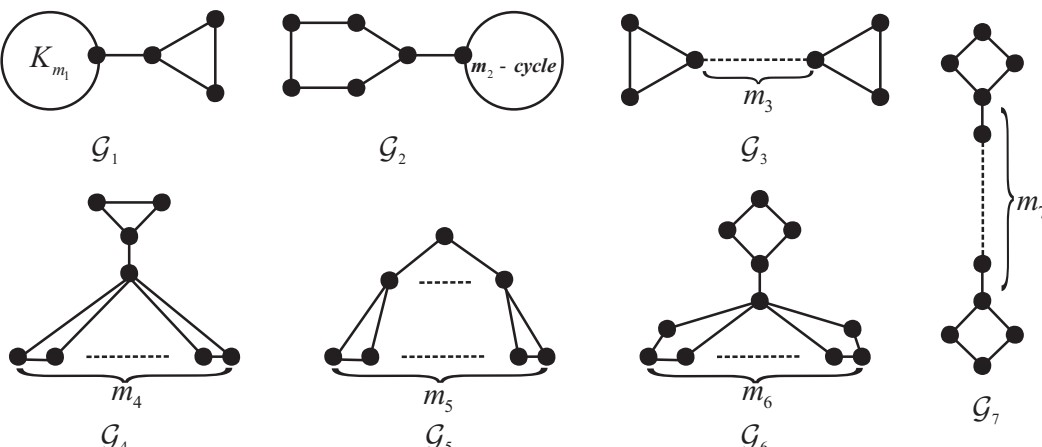

**Figure 2.** Some classes of graphs satisfies $1 = \kappa'(G) < \delta(G) = 2$.

**The necessity part of Theorem 4.** Let $S$ be a graph such that every connected $S$-free graph is $\kappa'(G) = \delta(G)$. Then $S$ is a common induced subgraph of all graphs in $\mathcal{G}_i, i = 1, \cdots, 7$.

Note that the common induced subgraph of the graphs in $\mathcal{G}_1$ and $\mathcal{G}_2$ is a path. Since the largest induced path of the graphs in $\mathcal{G}_1$ is $P_4$, $S$ must be an induced subgraph of $P_4$. This completes the proof of the necessity part of Theorem 4. $\square$

**The necessity part of Theorem 5.** Let $R$ and $S$ be not an induced subgraph of $P_4$ graphs such that every connected $\{R, S\}$-free graph is $\kappa'(G) = \delta(G)$. Then all graphs in $\mathcal{G}_i, i = 1, \cdots, 7$ should contain either $R$ or $S$ as an induced subgraph. Without loss of generality, we may assume that $R$ is a common induced subgraph of all graphs in $\mathcal{G}_1$. Note that all graphs in $\mathcal{G}_1$ contain no induced cycle with a length of at least 4 as an induced subgraph, so we need to consider the following four cases.

**Case 1.** $R$ contains a clique $K_t$ with $t \geq 4$.

Since for $i \in \{2, 3, 4, 5, 6, 7\}$, all graphs in $\mathcal{G}_i$ are $R$-free, they all should contain $S$ as an induced subgraph. Note that all graphs in $\mathcal{G}_2$ are $K_3$-free, and all graphs in $\mathcal{G}_3$ are $K_{1,3}$-free, so $S$ should be a path. Since the largest induced path of the graphs in $\mathcal{G}_4$ is $P_4$, $S$ should be an induced subgraph of $P_4$, a contradiction.

**Case 2.** $R$ does not contain the clique $K_t$ with $t \geq 4$, but contains two edge-disjoint $K_3$.

Since $R$ is a common induced subgraph of all graphs in $\mathcal{G}_1$, $R$ should be $H_1$. Since for $i \in \{2, 3, 5, 6, 7\}$, all graphs in $\mathcal{G}_i$ are $R$-free, they all should contain $S$ as an induced subgraph. Note that all graphs in $\mathcal{G}_2$ are $K_3$-free, and all graphs in $\mathcal{G}_3$ are $K_{1,3}$-free, so $S$ should be a path. Since the largest induced path of the graphs in $\mathcal{G}_5$ is $P_5$, $S$ should be an induced subgraph of $P_5$. So $\mathcal{H} = \{R, S\} \preceq \{H_1, P_5\}$.

**Case 3.** $R$ does not contain the clique $K_t$ with $t \geq 4$, but contains exactly one $K_3$.

Since $R$ is a common induced subgraph of all graphs in $\mathcal{G}_1$, $R$ should be an induced subgraph of $Z_2$. Since for $i \in \{2, 6, 7\}$, all graphs in $\mathcal{G}_i$ are $R$-free, they all should contain $S$ as an induced subgraph. Note that the common induced subgraph of all graphs in $\mathcal{G}_2$ and $\mathcal{G}_7$ is a tree with the maximum degree 3 or a path. If $S$ is a tree with the maximum degree 3, since the common induced tree with the maximum degree 3 of all graphs in $\mathcal{G}_6$ and $\mathcal{G}_7$ are $T_{1,1,3}$, $S$ should be an induced subgraph of $T_{1,1,3}$. So $\mathcal{H} = \{R, S\} \preceq \{Z_2, T_{1,1,3}\}$. If $S$ is a path. Since the largest induced path of the graphs in $\mathcal{G}_6$ is $P_6$, $S$ should be an induced subgraph of $P_6$. So $\mathcal{H} = \{R, S\} \preceq \{Z_2, P_6\}$.

**Case 4.** $R$ is a tree.

Since all graphs in $\mathcal{G}_1$ are $K_{1,3}$-free, $R$ should be a path. Note that the largest induced path of the graphs in $\mathcal{G}_1$ is $P_4$, so $R$ should be an induced subgraph of $P_4$, a contradiction.

From the proofs above, we have that $\mathcal{H} \preceq \{H_1, P_5\}$, $\mathcal{H} \preceq \{Z_2, P_6\}$, or $\mathcal{H} \preceq \{Z_2, T_{1,1,3}\}$. This completes the proof of the necessity part of Theorem 5. $\square$

## 4. The Sufficiency Part of Main Results

In this section, we provide the sufficiency proof of main results.

**The sufficiency part of Theorem 4.** Let $G$ be a connected $P_4$-free graph. Then $diam(G) \leq 2$. If $diam(G) = 1$, $G$ must be a complete graph and $\kappa'(G) = \delta(G) = n - 1$. If $diam(G) = 2$, by Theorem 3 (3), $\kappa'(G) = \delta(G)$. This completes the proof of the sufficiency part of Theorem 4. $\square$

**The sufficiency part of Theorem 5.** Let $G$ be a connected $\mathcal{H}$-free graph such that $\kappa'(G) < \delta(G)$, where $\mathcal{H} \preceq \{H_1, P_5\}, \{Z_2, P_6\}$, or $\{Z_2, T_{1,1,3}\}$. Then there must exist a minimum edge cut, say $M$, such that $|M| = \kappa'(G) < \delta(G)$. Let $G_1$ and $G_2$ be the components of $G - M$, and let $S_i = V(G_i) \cap V(M)$, $i \in \{1, 2\}$. Then $|S_i| \leq |M| = \kappa'(G) < \delta(G)$, say $|S_i| = s_i, i \in \{1, 2\}$.

**Claim 1.** For $i \in \{1, 2\}$, $V(G_i - S_i) \neq \emptyset$. Moreover, for any $x \in V(G_i - S_i)$, $N_G(x) \cap V(G_i - S_i) \neq \emptyset$.

**Proof.** We will count the number of edges of $G_i$ for $i \in \{1, 2\}$.

$$
\begin{aligned}
|E(G_i)| &= \frac{1}{2}\left( \sum_{x \in V(G_i)} d_G(x) - \kappa'(G) \right) \\
&\geq \frac{1}{2}\left( \delta(G)|V(G_i)| - \kappa'(G) \right) \\
&\geq \frac{1}{2}\left( \delta(G)s_i - \kappa'(G) \right) \\
&> \frac{1}{2}\kappa'(G)(s_i - 1) \\
&\geq \frac{1}{2}s_i(s_i - 1)
\end{aligned}
$$

Note that the complete graph $K_{s_i}$ has $\frac{1}{2}s_i(s_i - 1)$ edges. It means that $|V(G_i)| > s_i$, i.e., $V(G_i - S_i) \neq \emptyset$.

Moreover, for any $x \in V(G_i - S_i)$, since $d_G(x) \geq \delta(G) > \kappa'(G) \geq s_i$, $N_G(x) \cap V(G_i - S_i) \neq \emptyset$. This completes the proof of Claim 1. $\square$

Now we will distinguish the following two cases to complete our proof.

**Case 1.** $G$ contains a $P_4 = x_0 x_1 x_2 x_3$ with $x_0 \in V(G_1 - S_1), x_1 \in S_1, x_2 \in S_2$, and $x_3 \in V(G_2 - S_2)$.

**Subcase 1.1.** $\mathcal{H} \preceq \{H_1, P_5\}$.

By Claim 1, there exist two vertices $x_0' \in V(G_1 - S_1)$ and $x_3' \in V(G_2 - S_2)$ such that $x_0 x_0', x_3 x_3' \in E(G)$. Then $G[\{x_0', x_0, x_1, x_2, x_3, x_3'\}] \cong H_1$ (if $x_1 x_0', x_2 x_3' \in E(G)$), or $G[\{x_0', x_0, x_1, x_2, x_3\}] \cong P_5$ (if $x_1 x_0' \notin E(G)$), or $G[\{x_0, x_1, x_2, x_3, x_3'\}] \cong P_5$ (if $x_2 x_3' \notin E(G)$), a contradiction.

**Subcase 1.2.** $\mathcal{H} \preceq \{Z_2, P_6\}$.

By Claim 1, there exist two vertices $x_0' \in V(G_1 - S_1)$ and $x_3' \in V(G_2 - S_2)$ such that $x_0 x_0', x_3 x_3' \in E(G)$. Then $G[\{x_0', x_0, x_1, x_2, x_3, x_3'\}] \cong P_6$ (if $x_1 x_0', x_2 x_3' \notin E(G)$), or $G[\{x_0', x_0, x_1, x_2, x_3\}] \cong Z_2$ (if $x_1 x_0' \in E(G)$), or $G[\{x_0, x_1, x_2, x_3, x_3'\}] \cong Z_2$ (if $x_2 x_3' \in E(G)$), a contradiction.

**Subcase 1.3.** $\mathcal{H} \preceq \{Z_2, T_{1,1,3}\}$.

By Claim 1, $N_G(x_0) \cap V(G_1 - S_1) \neq \emptyset$ and $N_G(x_3) \cap V(G_2 - S_2) \neq \emptyset$.

Suppose that $|N_G(x_0) \cap V(G_1 - S_1)| \geq 2$ or $|N_G(x_3) \cap V(G_2 - S_2)| \geq 2$. Without loss of generality, we may assume that $|N_G(x_0) \cap V(G_1 - S_1)| \geq 2$, it means there exist two vertices $x_0', x_0'' \in V(G_1 - S_1)$ such that $x_0 x_0', x_0 x_0'' \in E(G)$.

Then

- $G[\{x_0', x_0'', x_0, x_1, x_2, x_3\}] \cong T_{1,1,3}$, if $x_0' x_0'', x_0' x_1, x_0'' x_1 \notin E(G)$;
- $G[\{x_0', x_0, x_1, x_2, x_3\}] \cong Z_2$, if $x_0' x_1 \in E(G)$;
- $G[\{x_0'', x_0, x_1, x_2, x_3\}] \cong Z_2$, if $x_0'' x_1 \in E(G)$;

- $G[\{x'_0, x''_0, x_0, x_1, x_2\}] \cong Z_2$, if $x''_0 x'_0 \in E(G)$ and $x'_0 x_1, x''_0 x_1 \notin E(G)$,

a contradiction.

Suppose that $N_G(x_0) \bigcap V(G_1 - S_1) = \{x'_0\}$ and $N_G(x_3) \bigcap V(G_2 - S_2) = \{x'_3\}$. Note that $N_G(x_0) \subseteq \{x'_0\} \bigcup S_1$ and $N_G(x_3) \subseteq \{x'_3\} \bigcup S_2$. Then $d_G(x_0) \le s_1 + 1$ and $d_G(x_3) \le s_2 + 1$. Since $d_G(x_0) \ge \delta(G) > \kappa'(G) \ge s_1$ and $d_G(x_3) \ge \delta(G) > \kappa'(G) \ge s_2$, $d_G(x_0) \ge s_1 + 1$ and $d_G(x_3) \ge s_2 + 1$. Therefore $d_G(x_0) = s_1 + 1$ and $d_G(x_3) = s_2 + 1$. It means that $N_G(x_0) = S_1 \bigcup \{x'_0\}$, $N_G(x_3) = S_2 \bigcup \{x'_3\}$, and $s_1 = s_2 = \kappa'(G)$. Since $|M| = \kappa'(G) = s_1 = s_2$, the each vertex in $S_1$ is just adjacent to exactly one vertex which is in $S_2$, and vice versa.

Suppose $s_1 \ge 2$. Then there exists a vertex $x'_1 \in S_1$ such that $x'_1 \neq x_1$. Therefore

- $G[\{x'_0, x_0, x'_1, x_1, x_2, x_3\}] \cong T_{1,1,3}$, if $x'_0 x'_1, x'_0 x_1, x'_1 x_1 \notin E(G)$);
- $G[\{x'_0, x_0, x_1, x_2, x_3\}] \cong Z_2$, if $x'_0 x_1 \in E(G)$;
- $G[\{x'_1, x_0, x_1, x_2, x_3\}] \cong Z_2$, if $x'_1 x_1 \in E(G)$;
- $G[\{x'_0, x'_1, x_0, x_1, x_2\}] \cong Z_2$, if $x'_0 x'_1 \in E(G)$ and $x'_0 x_1, x'_1 x_1 \notin E(G)$,

a contradiction.

Suppose $s_1 = 1$. Then $s_2 = \kappa'(G) = 1$ and $\delta(G) = 2$.

Assume $d_G(x_1) \ge 3$. Then there exists a vertex $x'_1 \in V(G_1 - S_1)$, such that $x'_1 x_1 \in E(G)$ and $x'_1 \neq x_0$. Therefore $G[\{x_0, x'_1, x_1, x_2, x_3, x'_3\}] \cong T_{1,1,3}$ (if $x_0 x'_1, x'_3 x_2 \notin E(G)$), or $G[\{x_0, x'_1, x_1, x_2, x_3\}] \cong Z_2$ (if $x_0 x'_1 \in E(G)$), or $G[\{x'_3, x_3, x_2, x_1, x_0\}] \cong Z_2$ (if $x'_3 x_2 \in E(G)$), a contradiction.

Assume $d_G(x_1) = 2$. Then it means that $N_G(x_1) = \{x_0, x_2\}$ and $d_G(x) = d_{G_1}(x)$ for any $x \in V(G_1 - \{x_0, x_1\})$. Since $\delta(G) = 2$ and $d_{G_1 - S_1}(x_0) = 1$, there exist some vertices in $V(G_1 - S_1)$ such that their degree in $G$ are at least 3. Then we choose a vertex $y_0 \in V(G_1 - S_1)$, such that $d_G(y_0) \ge 3$ and $d_G(y_0, x_1)$ as small as possible. Let $P'$ be the shortest path between $x_1$ and $y_0$. Then all inner vertices of $P'$ should have degree two. Let $y_1, y_2 \in N_G(y)$ and $y_1, y_2 \notin V(P')$. Then $G[\{y_1, y_2, x_2, x_3\} \bigcup V(P')]$ contains an induced $T_{1,1,3}$ (if $y_1 y_2 \notin E(G)$), or $G[\{y_1, y_2, x_2\} \bigcup V(P')]$ contains an induced $Z_2$ (if $y_1 y_2 \in E(G)$), a contradiction.

**Case 2.** $G$ contains no $P_4 = x_0 x_1 x_2 x_3$ with $x_0 \in V(G_1 - S_1), x_1 \in S_1, x_2 \in S_2$, and $x_3 \in V(G_2 - S_2)$.

Let $S_i^1 = \{x \in S_i : N_G(x) \bigcap V(G_i - S_i) \neq \varnothing\}$, and $S_i^2 = S_i - S_i^1$ for $i = 1, 2$. Then $S_i^2 \neq \varnothing$ and $E(S_1^1, S_2^1) = \varnothing$. By the minimality of $M$ and the definition of $S_i$, $E(S_i^1, S_i^2), E(S_1^1, S_2^2), E(S_1^2, S_2^1) \neq \varnothing$. Now we choose a path $P_0$ between $x_1$ and $x_2$, such that $x_1 \in S_1^1$ and $x_2 \in S_2^2$, and the length of the path is as small as possible. Then $|V(P_0)| \ge 3$ and all inner vertices of $P_0$ must be in $S_i^2$. Let $x_0 \in V(G_1 - S_1)$ and $x_3 \in V(G_2 - S_2)$, such that $x_0 x_1, x_2 x_3 \in E(G)$. Then $G[V(P_0) \bigcup \{x_0, x_3\}]$ is an induced path with at least 5 vertices, say $P_1$.

**Subcase 2.1.** $\mathcal{H} \preceq \{H_1, P_5\}$.

$P_1$ is an induced path with at least 5 vertices, a contradiction.

**Subcase 2.2.** $\mathcal{H} \preceq \{Z_2, P_6\}$.

By Claim 1, there exists a vertex $x'_0 \in V(G_1 - S_1)$ such that $x_0 x'_0 \in E(G)$. Then $G[\{x'_0\} \bigcup V(P_1)]$ contains an induced $P_6$ (if $x_1 x'_0 \notin E(G)$), or an induced $Z_2$ (if $x_1 x'_0 \in E(G)$), a contradiction.

**Subcase 2.3.** $\mathcal{H} \preceq \{Z_2, T_{1,1,3}\}$.

By Claim 1 and $|S_1^1| < s_1 < \delta(G)$, there exist two vertices $x'_0, x''_0 \in V(G_1 - S_1)$ such that $x_0 x'_0, x_0 x''_0 \in E(G)$. Then $G[\{x'_0, x''_0\} \bigcup V(P_1)]$ contains an induced $T_{1,1,3}$ (if $x'_0 x''_0, x'_0 x_1, x''_0 x_1 \notin E(G)$), or an induced $Z_2$ (if $x''_0 x'_0 \in E(G)$ and $x'_0 x_1, x''_0 x_1 \notin E(G)$), or an induced $Z_2$ (if $x'_0 x_1 \in E(G)$ or $x''_0 x_1 \in E(G)$), a contradiction.

This completes the proof of the sufficiency part of Theorem 5. □

## 5. Concluding Remark

In this paper, we give a complete characterization of all pairs $\{R, S\}$ of graphs such that every connected $\{R, S\}$-free graph has the same edge-connectivity and minimum degree. All graphs in Figure 2 have edge-connectivity one; we also can construct some graphs for arbitrarily large edge-connectivity to show that Theorem 4 also hold. But for

forbidden pairs $\mathcal{H} = \{R, S\}$, we do not have enough graphs to see that whether we could get wider forbidden pairs to guarantee the graphs having the same edge-connectivity and minimum degree, when we increase the edge-connectivity.

In fact, we obtain Theorem 6 which is more of a general case that deals with not only the relationship between edge-connectivity and minimum degree but also any parameters when they have the recurrence relation, while the content in [11] deals with the properties of graphs.

**Author Contributions:** Conceptualization, L.X.; methodology, Z.H. and J.D.; writing—original draft preparation, J.D.; writing—review and editing, L.X. and Z.H. All authors have read and agreed to the published version of the manuscript.

**Funding:** This research is supported by Natural Science Foundation of China (Nos. 11871099 (J.D.; L.X.), 11861069 (Z.H.), 12131013 (L.X.)).

**Acknowledgments:** The authors thank the two referees and the editor for the nice suggestion which makes the improvement of the presentation.

**Conflicts of Interest:** The authors declare no conflict of interest. The funders had no role in the design of the study; in the collection, analyses, or interpretation of data; in the writing of the manuscript, or in the decision to publish the results.

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
