# Peer review of "Characterizing Forbidden Pairs for the Edge-Connectivity of a Connected Graph to Be Its Minimum Degree"

_axioms, doi:10.3390/axioms11050219_

Round 1

Reviewer 1 Report

In this paper, the authors characterize all pairs of graphs {G,H} such that the edge-connectivity of every connected {G, H}-free graph is equal to the minimum degree of the same graph.

In the opinion of this reviewer, the paper is suitable for publication subject to a minor revision. Here are a few points to aid the readability of the article:

1) Lines 18-19 and 22-23, perhaps the sentences should read "every H_2-free graph is also H_1-free", rather than vice-versa.

2) Line 50: "one of the following" instead of "the one of following".

3) Lines 173-200: Perhaps this part of the proof should be presented in a neater manner.

4) Line 223: "characterization" instead of "characterzation".

Reviewer 2 Report

I really like this paper.

No need to change anything, but I would suggest:

- introduction is too long, and there are to many theorems, even one with proof. Maybe to split Introduction it into two sections: 0. Introductions and 1. Theoretical background  
- I do not like names of sections like they are for sections 1 and 2. Please, think about renaming.  Maybe to have one section MAIN RESULT, and that existing sections be subsections?
- Concluding remark is nice and clear

- Figures are nice and clear.
